



# Controls on the distribution of the soil organic matter in mountain permafrost regions on the north Qinghai-Tibet Plateau

Cuicui Mu, Tingjun Zhang, Xiankai Zhang, Hong Guo, Bin Cao, Lili Li, Hang Su, Xiaoqing Peng

Key Laboratory of Western China's Environmental Systems(Ministry of Education), College of Earth and Environmental Sciences, Lanzhou University, Lanzhou, 730000, China

*Correspondence to*: Tingjun Zhang (tjzhang@lzu.edu.cn)





**Abstract.** It has been known a large amount of soil organic carbon (SOC) have been accumulated over thousands of years and stored at considerable depths in permafrost regions, which could extent down tens of meters. Although the vegetation plays an important role in the distribution of SOC in upper 1 or 2 m soils, little is known about the determines of the organic carbon pools below these depths . We hypothesized that the SOM distribution and its chemical characteristics for different

depths were determined by vegetation types and soil texture in mountain permafrost. To test the hypothesis, ten boreholes which were about 20 m depth under alpine swamp meadow (ASM), alpine meadow (AM) and alpine steppe (AS) were drilled in the permafrost regions on the northern Qinghai Tibetan Plateau. The results showed that the SOC stocks were highest over ASM, and lowest over AS for different depths. The soil textures were mainly silt loam over ASM, while varied with sandy loam, silt loam, and sand in AM. All the soils with higher fine-fractions have higher SOC contents than that in

coarse soils. Meanwhile, the C/N ratios and carbon isotopes suggested that the SOC pools accompanied with fine-fractions soils under swamp meadow are more decomposable than those of coarse soils. Our results suggest and both the SOC stocks distribution and the chemical nature of organic matter are determined by the soil texture and vegetation types, and this rule is applicable for SOC distribution for the 20 m depth in mountain permafrost regions.

**Keywords:** mountain permafrost, soil texture, vegetation types, soil organic matter chemistry, C:N ratio, stable carbon isotopes

## 1 Introduction

Soil organic carbon (SOC) stored in permafrost regions will become accessible to microbial decomposition and may provide

a major source of greenhouse gases (Lee et al., 2012; Paré and Bedard‑Haughn, 2013). In addition to the large pools of SOC in circum-Arctic regions, there are also high stocks of SOC in mountain permafrost regions (Hoffmann et al., 2014a; Hoffmann et al., 2014b). The heterogeneities (topography, soil hydrothermal conditions, aspect and vegetation type) result in a large variability of soil properties and thus large uncertainties in the SOC distribution (Hoffmann et al., 2014a).

In mountain permafrost regions, it has been showed that many factors such as precipitation (Yang et al., 2008),

permafrost, pedogenesis (Baumann et al., 2009), vegetation types (Wu et al., 2012), below ground biomass (Liu et al., 2012), soil moisture, and soil texture (Wu et al., 2016) may have effects on the SOC distribution for the upper 0-2 m soil layers, while little is known about the controlling factors for the SOC pools below these depths. In permafrost regions, the deep SOC pools are vulnerable to mobilization following global warming (Bockheim and Hinkel, 2007). A recent study showed that deep SOC in permafrost regions may greatly affect the carbon balance since these carbon pools can have great soil

respiration that are not compensated by inputs (Koven et al., 2015).

The chemical composition of organic matter is an important variable with respect to potential bioavailability in a warming future (Mueller et al., 2015). In circum-Arctic regions, it has been demonstrated that deep soil carbon in permafrost





regions also has high microbial availability (Schuur et al., 2015). The C/N ratio in soils provides an important information in how soils respond to environmental changes (Callesen et al., 2007) because highly decomposed organic matter has low C/N

ratios (Werth and Kuzyakov, 2010). Stable carbon isotope analysis is an useful tool for gaining insight into biogeochemical processes involved in SOC degradation (Jones et al., 2010). $^{12}$C is preferentially used by decomposers, which may lead to $^{13}$C enrichment in the remaining SOC (Natelhoffer and Fry, 1988), consequently resulting in easily decomposable substances having less negative $\delta^{13}$C values. Therefore, the C/N ratios, and $\delta^{13}$C values organic matter could be used to reveal the potential microbial decomposition of organic matter in permafrost regions.

As a typical mountain permafrost region, the Qinghai-Tibetan Plateau (QTP) is the largest middle and low latitude permafrost area. The QTP is extremely sensitive to global warming since the rate of climate warming could be amplified in high-mountain regions (Kang et al., 2010). For the permafrost regions on the QTP, most ground temperatures are slightly below 0 ℃ (Zhao et al., 2010), so small increases in air temperature will cause rapid soil organic matter degradation (Ping et al., 2015). The SOC storage in the QTP is approximately 160 Pg C for the $150 \times 10^4$ km$^2$ of the 0-25 m soils (Mu et al., 2015).

Of that, 132.3±76.8 Pg C is stored in soils below 3 m. There are several reports to the distribution and SOM chemistry in the permafrost regions on the QTP (Hu et al., 2014; Wu et al., 2014), while little is known about the distribution of deep SOC and its chemistry.

The SOC pool in deep soils was accumulated thousands of years (Mu et al., 2014; Schuur et al., 2009). In many soils, the soil texture is related positively to SOC contents (Schimel et al., 1994), and also has important effects on soil hydro-thermal

properties (Sugimoto et al., 2003). Moreover, soil texture also relates to vegetation types, soil moisture and even active layer thickness of permafrost (Wu et al., 2016). Based on the above understanding, we propose the following hypotheses: (1) the distribution of SOC in the soils including surface and deep layers is influenced by vegetation types and soil texture in the QTP; (2) there are significant differences in the characteristics of SOC under different vegetation types and different soil texture classes. These hypotheses were tested using soils from the northern QTP by analyzing the contents, C/N ratios and

stable carbon isotopes.

## 2. Materials and methods

### 2.1 Study area

The study area is located in the Qilian Mountains at the upper reach of the Heihe River Basin in the northern QTP (Fig. 1).

The area belongs to a region of inland-river in a semi-arid area in China and was characterized by prevailing westerlies (Wang et al., 2013). The annual air temperature of the basin ranges is less than 2 ℃. The annual precipitation is about 400 m, and the mean annual evaporation is 1,080 mm (Peng et al., 2013).

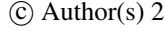



The distribution of permafrost in the Heihe River Basin is mainly controlled by elevation, and the lower limit of the elevation of permafrost was approximately 3,650 m (Wang et al., 2013). There are three main vegetation types in the study
area: alpine swamp meadow, alpine meadow and alpine steppe, and the dominant plants are *Kobresia tibetic*, *Kobresia pygmaea*, and *Kobresia humilis*. The geological stratigraphy of the study area is quaternary.

### 2.2 Field sampling and monitoring

The SOC pools in permafrost regions were usually calculated according to distribution of vegetation types (Mu et al., 2015; Ping et al., 2008). In the permafrost regions on the QTP, there are mainly three vegetation types, i.e., alpine steppe (AS),
alpine meadow (AM), and alpine swamp meadow (ASM) (Ding et al., 2016; Hu et al., 2014; Wu et al., 2016). Therefore, ten deep boreholes, including AS, AM, and ASM were drilled at elevations of 3,615-4,138 m in 2011-2014 (Fig. 1). The soil parent materials for PT sites (PT4, 5, 6, 7, 9, 10, 11, 12) were alluvium, and these sites were located in a mountain basin with a gradually slope. The parent materials for EB1 and EB2 sites were colluvium, located in a mountain valley. The geographic information of these boreholes is shown in Table 1. The vegetation, permafrost and basic soil properties were shown in
Table 2. Overall, the vegetation type of PT4, PT5, PT6 and PT7 was AM, that of PT9, EB1, and EB2 was ASM, and that of PT10, PT11 and PT12 was AS. The borehole site of PT11 was located in a seasonally frozen ground area, and the other borehole sites were located in permafrost areas. The sites of PT10, PT11, and PT12 have the same aspect, slope, topography, elevation and well drainage conditions, and are considered as the boundaries of permafrost and non-permafrost. The soils in the study area were largely alkaline with pH values of 8.53-8.84, with the exception for EB1 and EB2. The conductivities of
the soil suspensions ranged from 0.95 to 1.33ms cm$^{-1}$ (Table 2).

The depths of the drilled boreholes were approximately 20 m. The collected core diameter was about 15 cm. The depths of samples collected at PT6, PT9, EB1 and EB2 were 9.0, 7.0, 6.0 and 5.0 m because below these depths there were rock layers. For PT12, the soil samples in the upper 2 m were not available because of the high gravel contents in this layer. For all other sites, each 30-40-cm-long drilled core was recovered, photographed, wrapped, labeled on the both ends of the core. These
cores were stored in a freezer at -20 ℃. Upon returning to the laboratory at Lanzhou University, the samples were transferred to an ultralow-temperature freezer.

### 2.3 Laboratory analyses

#### 2.3.1. Basic soil analyses

Soil bulk density was determined by calculating the volume of a section from the frozen core before drying the section and
determining its mass. Total water content was determined by drying the soils at 105 ℃ for 8 h and measuring the soil weight before and after drying. The pH values and conductivities of the soil suspensions (1:2.5 soil:water ratio) were measured with a pH meter and conductivity meter. The percent by weight of rock fragment (>2mm) was calculated with the oven-dried



samples. The soils particle distribution was separated into three fractions: clay (< 2 μm), silt (2- 50 μm), and sand (50 μm – 2 mm) by a laser diffraction instrument (Malvern Mastersizer 2000, Malvern, UK).

The SOC and total nitrogen (TN) of pulverized homogenized samples were quantified by dry combustion using a Vario EL elemental analyzer (Elemental, Hanau, Germany). The SOC contents for EB1 and EB2 sites were cited from previous study(Mu et al., 2015). To measure the SOC, 0.5 g dry soil samples were pretreated with HCl (10 mL, 1 mol L$^{-1}$) for 24 h to remove carbonate. The C/N ratios were calculated using the mass ratio between SOC and TN (Ping et al., 2015).

### 2.3.2. Isotope analyses

The stable carbon isotopes of EB1 and EB2 were cited from previous study (Mu et al., 2014). The stable carbon isotopes of the SOC of other samples were analyzed using an OI Analytical Analyzer with an analytical precision of ±0.2‰ (Picarro, California, USA). The samples were treated with hydrochloric acid to remove inorganic carbon prior to analysis. The results are based on the mean of three replicates for each sample and are expressed as δ-values relative to the VPDB δ$^{13}$C standard. The δ-values are defined as follows:

$$\delta^{13}C\text{‰} = \left[ \left( R_{sample} / R_{standard} \right) - 1 \right] \times 1000$$

where R$_{sample}$ and R$_{standard}$ are the $^{13}$C/$^{12}$C ratios of the samples and standard, respectively.

### 2.3.3. Statistical analyses

The data presented in this study are the average values. The linear regression was performed using ANOVA.

## 3. Results

### 3.1 Distribution of soil organic carbon and C:N ratios

The distribution of SOC densities varied among different vegetation types and depth (Fig. 2). For the sites of ASM (PT9, EB1 and EB2), the SOC densities were much higher than those of AM, although there was a decreasing trend along with depth at EB2. The mean SOC densities for the sites ranged from 0.4 to 22.4 kg m$^{-3}$, with the highest value appeared at ASM.

The lowest SOC densities were recorded at the sites in AS sites (PT10, PT11 and PT12), with mean values of less than 1.0 kg m$^{-3}$.

For all the measured samples, the C/N ratios ranged from 2.26 to 73.04. The distribution of C/N ratios with depth followed a similar trend to the SOC densities at the sites of PT4-7 (Fig. 2). The average values of C:N ratios in permafrost boreholes were 19.98, 17.65, 13.61, 13.44 for the sites of PT4, PT5, PT6, PT7, respectively. For the sites of ASM, the C:N

ratios for PT9, EB1, and EB2 were 11.03, 7.59, and 6.45.



### 3.2 Relationship between soil organic carbon and C/N ratios and stable carbon isotopes

For the samples at different depths, the C/N ratio and SOC content had a weak positive relationship for the ASM sites ($r^2$=0.028, $p<0.05$, Fig. 3a), while had a higher correlation for AS and AM sites ($r^2$=0.522, $p<0.001$, Fig. 3b).

The stable carbon isotope ($\delta^{13}$C‰) largely varied from -29.9‰ to -22.8‰ in the samples with an exception for some deep samples collected at EB2 (below 3.5 m, and the soil texture belongs to sand). Overall, there were significant negative relationships between C/N ratios and $\delta^{13}$C‰ values both for ASM sites (Fig. 3c) and AS/AM sites (Fig. 3d).

### 3.3 Relationship between SOC and environmental factors

There was no significant correlation between the SOC content and total water content for ASM sites (Fig. 4a), while a significantly positive relationship was found in AS and AM sites (Fig. 4b). Significant negative relationships were found between SOC content and bulk density for the two groups (Fig. 4c and Fig. 4d).

According to the soil particle distribution, the soils from the sampling sites belong to the texture of sand, loamy sand, sandy loam, and silt loam. The loamy sands were only recorded in the upper 2 m layers at PT4, and the silt loams were recorded in

the most soils of ASM sites (PT9, EB1, and EB2), AM site of PT6, and upper 10 cm layer of PT5. It is clearly that the silt loams of ASM sites had the highest SOC densities, followed by the sandy loams and silt loams of AM sites. The sands under AS sites had higher SOC densities than that of ASM, while the lowest SOC densities appeared in the sands of AS sites (Fig.5).

For all the samples, the SOC contents were significantly correlated to the clay contents ($R^2$=0.232, $p<0.001$, n=158). The

correlation between SOC and clay in ASM sites ($R^2$=0.562, $p<0.001$) were higher than those of AM and AS sites ($R^2$=0.139, $p<0.001$) (Fig. 6).

According to the relationship between the SOC contents and depth, total water content, bulk density, and clay content, the best regression models are displayed as follows:

$$\text{SOC}(\%) = 0.092 \times \text{Moisture} - 0.214 \times \text{Depth} - 0.201 \times \text{Gravel} + 0.672 \times \text{Clay} \ (R^2 = 0.566, p < 0.001, n = 158) \ .$$

### 4. Discussion


The SOC densities of the three vegetation types are similar with of previous reports in the eastern (Ding et al., 2016; Hu et al., 2014; Yang et al., 2008), middle (Baumann et al., 2009; Shang et al., 2016), and western (Wu et al., 2012; Wu et al., 2016) part of the QTP, although these results mainly focused on the SOC within 2 m or 3 m depths. For deep layers, the SOC densities are also similar with the results from the boreholes which distributed along a latitude gradient (Mu et al.,

2015). Therefore, the sampling area can be considered as representative of SOC distribution for the QTP. The SOC densities



of the vegetation types of AM sites (PT4, PT5, PT6, PT7) were higher than those of AS sites (PT10, PT11 and PT12), while lower than those of ASM sites (PT9, EB1, and EB2). This finding was in agreement with the previous results that vegetation types affect the SOC contents (Jobbágy and Jackson, 2000; Wu et al., 2012). It is worth mentioning that this pattern was not only limited to the upper 2 or 3 m layers (which were usually studied in previous reports) but also extended to the deep

permafrost layers, which could reach to 5 m depth or more (PT6, PT9, EB1, EB2) and even about 20 m depth (PT4, PT5, PT7). In Arctic regions, yedoma deposits and river deltas had been recognized have thick soil layers with high organic carbon (Hugelius et al., 2014; Tarnocai et al., 2009). Our results showed that the vegetation types potentially have great effects on SOC distribution in deep layers in mountain permafrost areas on the north QTP.

For the same vegetation type of AM, the SOC densities in the active layer at sites PT4, PT5, PT6 and PT7 had a wide

range from 4.77 to 11.12 kg m$^{-3}$, which could be related to topography, elevation, aspect and slope (Hoffmann et al., 2014a; Thompson and Kolka, 2005). Although the SOC densities at the AS sites (PT10, PT11 and PT12) showed a decreasing trends with depth in the active and permafrost layers, the SOC densities in the permafrost boreholes (PT4, PT5, PT6, PT7, PT9, EB1, and EB2) were still high in some deep layers. This shows that low temperature contributes to the preservation of SOM (Zimov et al., 2006). The results indicate that the vegetation types and permafrost in mountain permafrost regions have

a significant effect on the vertical distribution of SOC both in active and permafrost layers.

The C/N ratio showed significantly positive correlation to SOC contents when combined all the samples at different layers. The significant positive relationship between the C/N ratios and SOC contents (Fig. 3a and 3b) was consistent with the notion that high C/N ratios reflect a better preservation of SOM for the different depths (Andersson et al., 2012). This was corroborated by the relationships between soil water and SOC contents since higher soil water content will decrease the

decomposition of organic carbon (Schlesinger and Andrews, 2000). The mean values of C/N ratios at ASM sites were much lower than those of AM and AS sites. According to the global database of soils, the C/N ratios for tundra were about 18 (Eswaran et al., 1993; Post et al., 1985). Therefore, the relative lower C/N ratios in the present study implies that SOC stored in the ASM will be easily decomposable and thus has great potential to produce greenhouse gases in the future.

The δ$^{13}$C values largely ranged from -29.6‰ to -22.8‰ in the study area, which is within the range generally associated

with C3 peat-forming plants. The δ$^{13}$C‰ negatively correlated to SOC contents suggests that the lower microbial decomposition is one of the mechanisms for the accumulation of the SOC. For the sites of AM and AS in this study with good drainage, it was could be inferred that aerobic conditions that favor the selective loss of $^{12}$C (Alewell et al., 2011) and thus the C/N ratio was expected has a negative relationship with carbon isotope for the samples at different depths (Fig 3c, 3d). It worth mentioning that the δ$^{13}$C‰ values of the soils below 3.5 m at EB2 were much higher than other samples of

ASM sites. This could be explained by that the texture of sand greatly promoted the decomposition of SOM during the accumulation process of these carbon pools, and/or a change of vegetation from C4 plants to C3 plants at the depths of 3.5 m (Mu et al., 2014).

In our study, the clay content significantly correlated to the SOC contents. This is in agreement with the reports for the upper 2 m layers (Wu et al., 2016). This could be explained as that fine particles tend to stabilize and retain more organic





matter than coarser particle (Gregorich et al., 1994). In addition, fine particles have higher water holding capacity (Gómez-Plaza et al., 2001). In the three ASM sites, these factors were not significantly relationship between the SOC contents and gravel content, depth, and moisture, and there were even no gravels in the soils at EB1 and EB2. . Therefore, although these factors were appeared as independent factors for the SOC contents in the linear regression model, these factors largely reflected the SOC patterns in AS and AM sites.


## 5. Conclusions

The SOC distribution and the chemical characteristics of SOM under different vegetation types in mountain permafrost regions were investigated on the northern QTP. The vegetation types have an important effect on the SOC distribution in different soil layers which could reach to 20 m depth. Soil texture is another determinants for the SOC pools both in the

active layer and permafrost, and the fine soil particle is favor to the accumulation of SOC. The higher C/N ratios and lower stable carbon isotopes suggested the chemical nature of organic matter was also affected by vegetation types and soil texture, and the SOC pools accompanied with fine-fractions soils under swamp meadow are more decomposable than those of coarse soils. The result improves our understanding of SOC distribution, as well as provides new insights into evaluation the interaction the SOC pools and global warming in future.

**Acknowledgements**

This work was supported by the National Key Scientific Research Project (Grant 2013CBA01802), and the National Natural Science Foundation of China (Grants 91325202, 41330634). This work was also supported in part by the Open Foundations of State Key Laboratory of Frozen Soil Engineering (Grant SKLFSE201408), the State Key Laboratory of Cryospheric Sciences (Grant SKLCS-OP-2014-08) and the Fundamental Research Funds for the Central Universities (Grant lzujbky-

215 2015-123).

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





Table 1 Geomorphic and permafrost conditions for the sampling sites

| Site | Longitude (°) | Latitude (°) | Altitude (m) | Active layer thickness (m) | Aspect | Slope (°) | Geomorphic region | Drainage class | Borehole depth (m) | Soil depths (m) |
|------|------|------|------|------|------|------|------|------|------|------|
| PT4 | 98.946 | 38.833 | 3770 | 3.63 | flat | flat | piedmont plain | P | 90 m | 32.5 m |
| PT5 | 99.026 | 38.806 | 3692 | 3.42 | flat | flat | piedmont plain | P | 20 m | >20m (clay for the soils below 16 m) |
| PT6 | 98.963 | 38.955 | 4159 | 2.40 | southeast | 2 | piedmont slope | P | 50 m | 9 m |
| PT7 | 98.963 | 38.903 | 3970 | 2.41 | northeast | 1.5 | piedmont plain | P | 36 m | 25.5 m |
| PT9 | 98.950 | 38.627 | 4138 | 1.63 | eastern | 2 | piedmont slope | VP | 160 m | 7 m |
| PT10 | 99.068 | 38.789 | 3681 | 4.85 | flat | flat | piedmont plain | SE | 20 m | >20 m |
| PT11 | 99.0680 | 38.789 | 3680 | SFG | flat | flat | piedmont plain | SE | 20 m | >20 m |
| PT12 | 100.920 | 38.788 | 3680 | 5.75 | flat | flat | piedmont plain | SE | 20 m | >20 m |
| EB1 | 100.916 | 37.998 | 3700 | 1.2 | northwest | 1.2 | piedmont slope | VP | 20 m | 8 m (high ice contents for 6-8 m) |
| EB2 | 100.907 | 38.003 | 3615 | 1.3 | northwest | 2.5 | piedmont slope | VP | 11.7 m | 8 m (high gravel contents for 5-8 m) |



Table 2 Vegetation and selected soil properties in upper reach of the Heihe River basin

| Site | Vegetation types | Vegetation cover% | Dominant species | Soil pH | Conductivity( ms cm$^{-1}$) | Conductivity( ms cm$^{-1}$) |
|---|---|---|---|---|---|---|
| PT4 | AM | 94 | *Kobresia* | 8.53 | 1.11 | 1.11 |
| PT5 | AM | 90 | *pygmaea* C. B. Clarke | 8.84 | 0.95 | 0.95 |
| PT6 | AM | 80 | *Ajania tibetica* | 8.64 | 1.05 | 1.05 |
| PT7 | AM | 85 | *Rhodiola subopposita* | 8.70 | 1.09 | 1.09 |
| PT9 | ASM | 96 | *K.tibetica Maxim* | 8.65 | 1.12 | 1.12 |
| PT10 | AS | 87 | *K. humilis* | 8.76 | 1.14 | 1.14 |
| PT11 | AS | 78 | (C. A. Mey. ) | 8.61 | 1.03 | 1.03 |
| PT12 | AS | 75 | Serg. | 8.80 | 1.83 | 1.83 |
| EB1 | ASM | 95 | *K.tibetica Maxim* | 6.58 | 1.02 | 1.02 |
| EB2 | ASM | 95 | *K.tibetica Maxim* | 7.44 | 1.33 | 1.33 |

ASM: alpine swamp meadow; AM: alpine meadow; AS: alpine steppe



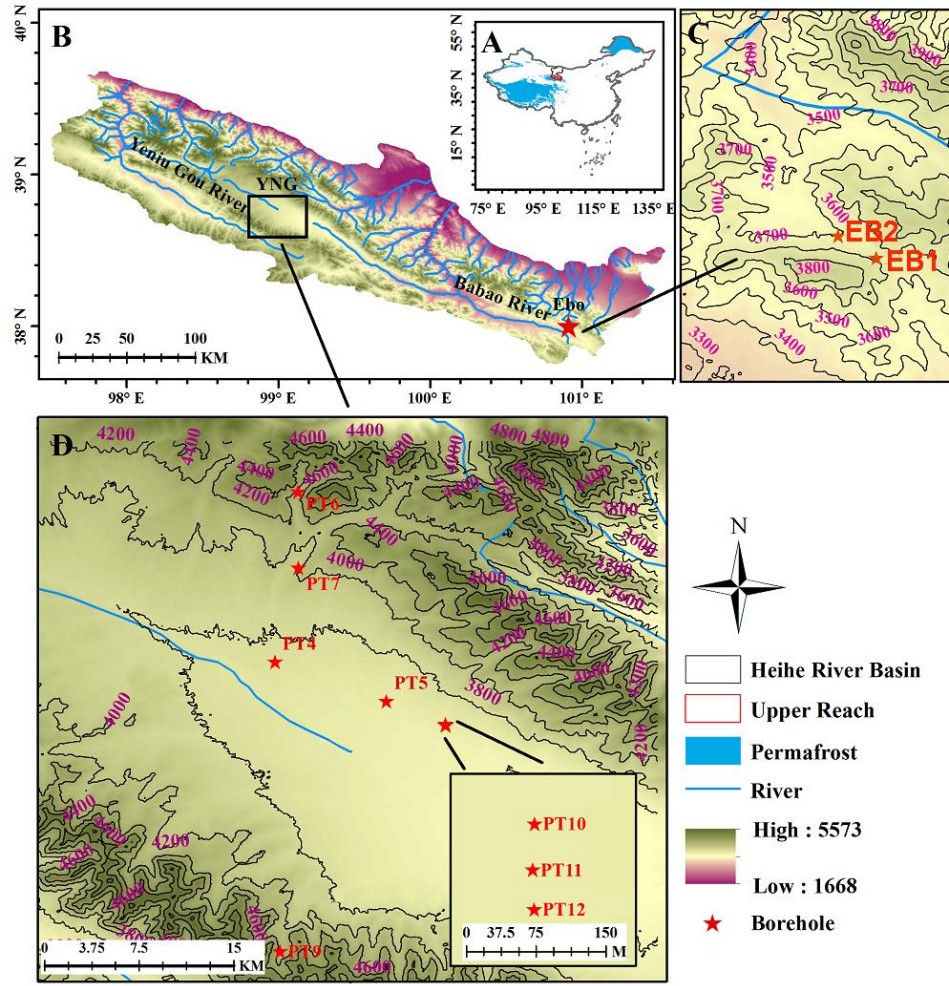

**Figure 1 Location of permafrost borehole sites in upper reach of the Heihe River, northern Qinghai Tibetan Plateau**





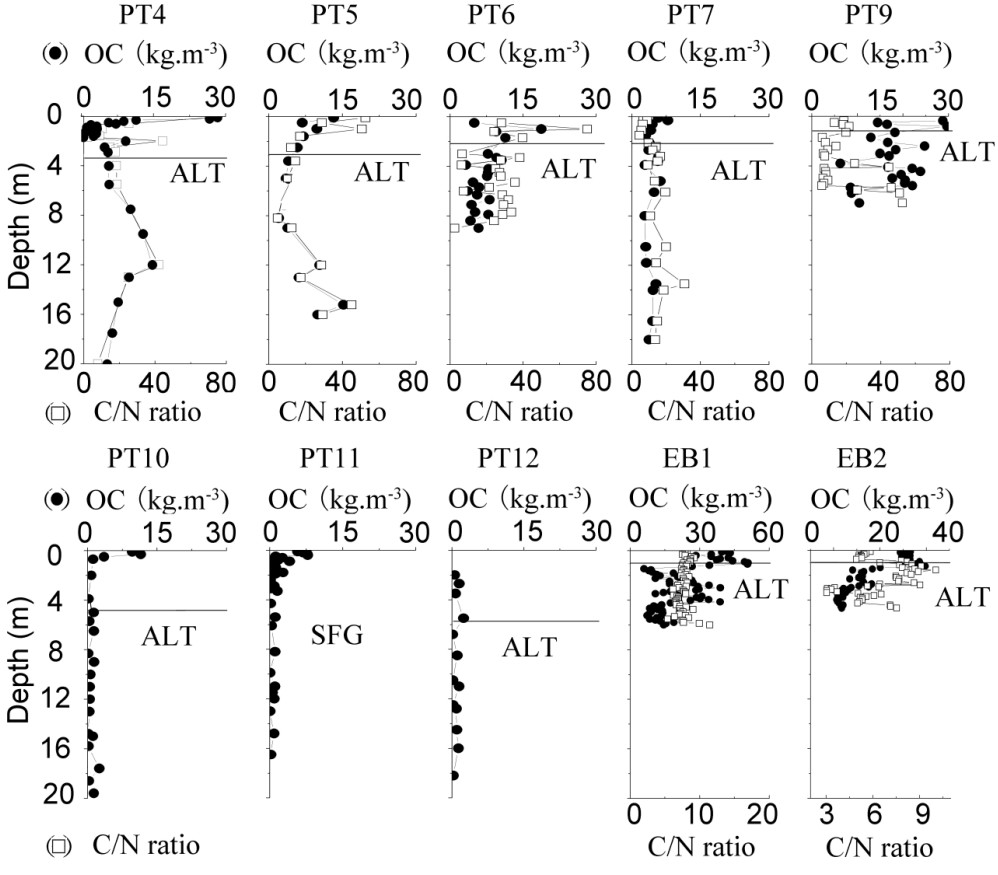

**Figure 2 Distribution of soil organic carbon densities and C/N ratios with depth at all sites (the nitrogen data were not available for PT10, PT11, and PT12) (OC, soil organic carbon; ALT, active layer thickness)**



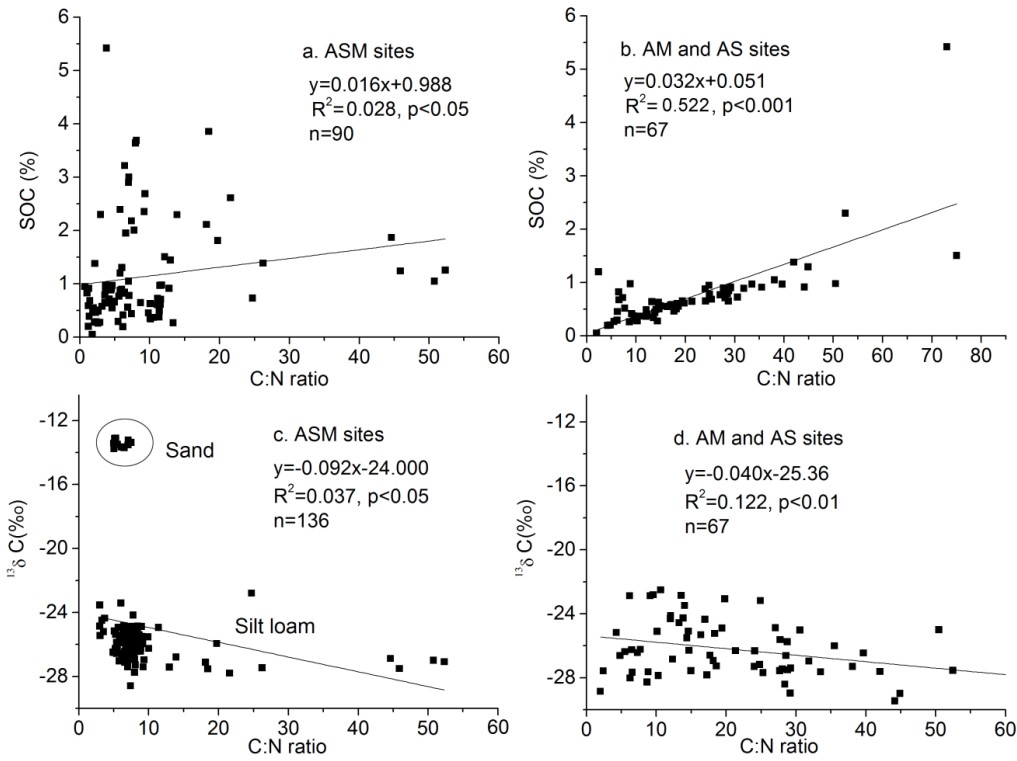

**Figure 3 Correlations between SOC content, stable carbon isotope (δ¹³C‰) and C:N ratio for ASM, AS and AM**

**sites (ASM: alpine swamp meadow; AS: Alpine steppe; AM: Alpine meadow)**





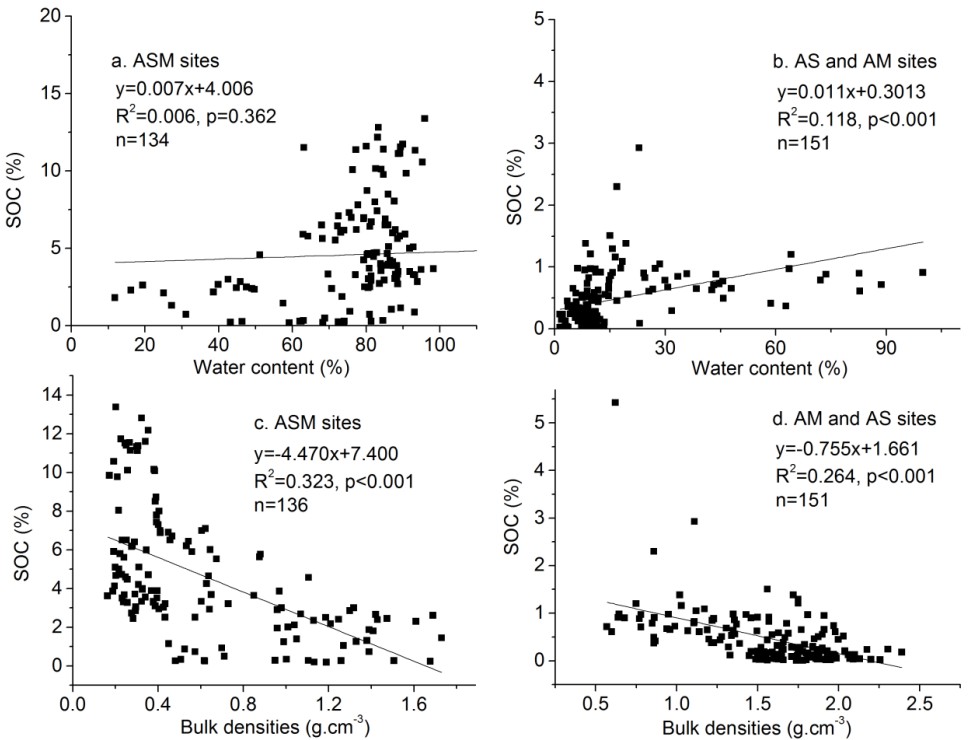

**Figure 4 Correlations between SOC and water content, bulk density for ASM, AS, and AM sites (ASM: alpine swamp meadow; AS: Alpine steppe; AM: Alpine meadow)**



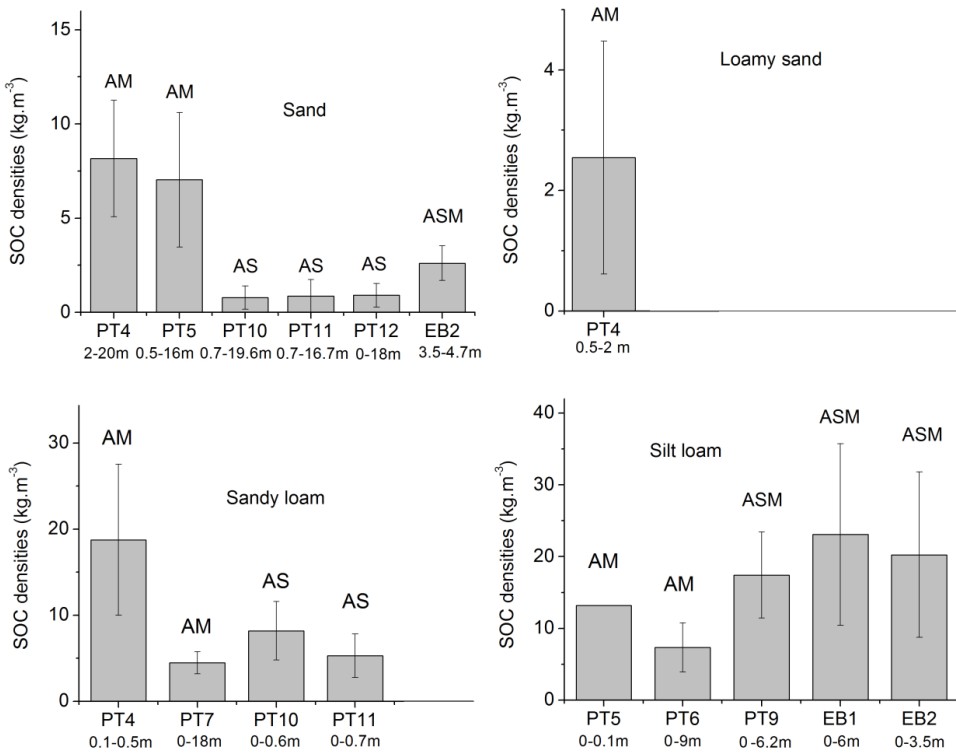

**Figure 5 SOC densities in different soil textures at different depths under different vegetation types (ASM: alpine swamp meadow; AS: Alpine steppe; AM: Alpine meadow)**



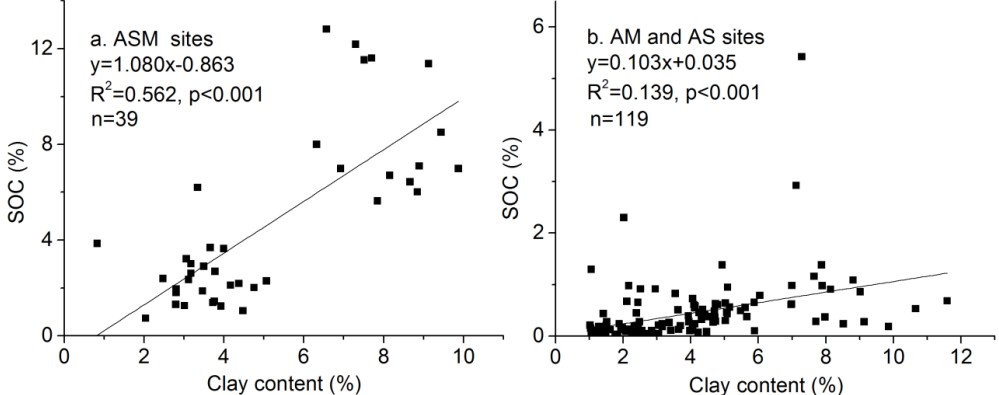

**Figure 6 Correlations between SOC and clay contents for ASM, AS and AM sites (ASM: alpine swamp meadow; AS: Alpine steppe; AM: Alpine meadow)**