# Peer review of "Controls on the distribution of the soil organic matter in mountain permafrost regions on the north Qinghai-Tibet Plateau"

_The Cryosphere, 2016_

## Referee Comment (RC1) · Anonymous Referee #1 · 17 May 2016

Review of Mu et al. Manuscript General comments This manuscript is about the controls of SOM distribution in a mountain permafrost region of the QZP. The authors over simplified the factors controlling the carbon stores/density and distribution of SOC in deep strata in a permafrost environment. The discussion and conclusion ( as in lines 21-23 and lines 160-169). The conclusions may well apply to the active layer or 0-2 m as described in many other papers. The near surface SOM is biogenic, resulting from the biomass accumulation form the vegetation community. However, the SOC accumulated in the deep strata may, and often the result of the geomorphic processes such as erosion and sedimentation. Therefore the SOC stores in deep layers may not be controlled by vegetation as what is currently on the surface. Since the parent materials

of these soils or cores studied are of Quaternary age, the past climate, vegetation, and especially the mode of deposition are important to the SOC stores; as we consider the syngenetic nature of the soil development. The major problem for me to review this manuscript is the lack of original data. There is no tabulated data for each analyzed soil horizon or layer. Thus there is no way to tell if the values of pH, soil texture, conductivity, C-density, C/N ratio presented are from one particular section/horizon or the average of the whole sampling depth. The authors are responsible should provide the original data as supplement that should include all the analyses as indicated in the Method section, and present the analytical data of each sampled layers of sections. The soils are likely syngenetic if the particle size distribution is more relatively uniform, or fluvial/erosion or sedimentation modification if there are contrasting soil textures. The %SOC correlates well with %clay because of physical protection, a function of surface area. But the fine soil particles are not limited to clay. There are several papers dealing with SOC contents in both the clay and silt fractions. Will there be any difference if the %silt is considered in the correlation? Soil drainage is mentioned in Table 1 but not discussed. The SOC content is controlled by vegetation community which is affected by drainage or soil water content due to soil texture and landform position. I recommend the manuscript be accepted upon major revision. Specific comments L. 18. "silt loam over ASM" change "over" to "in". L. 19-20. "higher fine-fractions" change to "higher finer textured fractions"; "coarse soils" change to "coarser-textured soils". L. 21. "more decomposable" or "more decomposed"? L. 29. Insert "more" after "become". Insert "due to climate warming" after "decomposition". L. 34. Delete "permafrost". L. 38. Citation "(Bockheim and Hinkel, 2007)" is not the proper reference. In their paper, the deepest soil horizon was sampled at 161 cm. This can hardly be called "deep carbon". For the vulnerability of deep carbon, refer to the papers by Schuur et al. and Zimov et al. L. 46-48. "consequently resulting in easily decomposable substances". It is because the original substrate was easily decomposable thus resulting in less negative delta 12C values. So consequently resulting in highly decomposed substances as indicated by lower C/N values. L. 60. "soil texture also relates to vegetation types". Soil texture is one of the several factors affecting vegetation types. L. 64. Insert "carbon" between "the" and "contents". L. 70. Change "area" to "region". Last word "westerlies"? L. 71. Insert "mean" before "annual". L. 74. Change "main" to "major". L. 76. Capitalize "Quaternary"/ L. 80. Change "gradually" to "gradual". L. 86. Last word "largely" change to "strongly". Are the pH values and electric conductivity measured for only the surface soil (topsoils) or average values for the whole profile (down to the bottom of sampling? For definition for saline and alkaline soils, see http://www.nrcs.usda.gov/Internet/FSE_DOCUMENTS/nrcs142p2_052523.pdf Reaction of EB1 is neutral and EB2 is slightly alkaline. L. 99. Delete "values" and change "conductivities" to singular. L. 100-103. Where is the data for soil texture and rock fragment content? L. 103. Insert "contents" after "(TN)". L. 120. The variation of vegetation type is limited to 2 Kobresia species. L. 122. The SOC density ranged from 0.4 to 22.4 kg m-3. Is the density of different soil horizon or this is the average of the whole soil profile? If so, then what is the carbon stores (kg m-2) of the active layer or 0-2 m and the whole profile? L. 128. Add "respectively" at the end of sentence. L. 141. Insert "class" after "texture". L. 151. "moisture" is not the proper word; water! The equation is poorly constructed. Use symbols; in line 150, add (D) after depth, add (W) after water content, add (Db) after bulk density, and "Cy" or other choice after clay content. L. 155. See general comments.

Tables Need footnote for the Drainage class in Table 1. Whay are there 2 columns of conductivity? Tables 1 and 2 should be combined and titled "Physiographic environment of the study sites in the Heihe River basin, Qinghai_Tibetan Plateau". Soil properties should be in another table. Besides pH, conductivity, the methods section also include SOC and water contents, C/N ratio, bulk density, particle size distribution, rock fragment content.
* * *

---

## Referee Comment (RC2) · Anonymous Referee #2 · 29 May 2016

This manuscript tried to clarify the main factors for affecting the SOC densities in the Tibetan Plateau based on dataset from the far north corner of the Tibetan Plateau, source region of Heihe River, Qilian Mountains. The dataset was valuable in such a data-absent region, specially, the dataset from deeper layer of soils. But the conclusions were general knowledge, which could be found in most related literatures.

The concerns and suggestions include: 1. It could be seen from Figure 1 that all the boreholes are located on the bottom of valley and the lower part (gentle) of the slopes. The sites even could not cover all the surface ecological and geological conditions of the study region. The dataset is so limited, just 10 boreholes to be considered as representatives of 3 types of ground surface conditions (AS, AM and ASM). Furthermore,

the sampling sites are located in the far northeast boundary of the Plateau, and the area of the study region accounts for less than ten thousandth of Tibetan Plateau. All the geologic, geomorphologic, geographical and climatic backgrounds are great different from the real plateau. I do think that the dataset collected in this region just can be representative of the local condition, even not as representative of Qilian Mountain Ranges, because the climatic conditions is also great different to the western part of the mountain range. It would be better if the title of the manuscripts revised as "Controls on the distribution of the soil organic matter in the Upper Reach of Heihe River, Qilian Mountains. 2. The results in section 3.1 and 3.2 are very general description for soil organic carbon, C:N ratios and stable carbon isotopes. The highest soil organic carbon density was found in boreholes under ASM, and the lowest was at AS. Similar results were reported in great amount of literatures by Wang, et al. and Wu et al., but there is no more new. 3. SOC in deeper soil layers should be affected more by paleo-climatic, ecological and geological background of the soil formation. The authors simply correlated SOC with the moisture content and texture (gravel and clay) of the soils. It would be better to add more information about soil formation history and discuss the controlling factors of SOC for different soil layers separately. 4. L160-168: it is a general knowledge that SOC is produced by photosynthesis of plants. There is without exception for organic carbon deposited in deep soil layers. Therefore, generally speaking, the better in the vegetation, and the higher in SOC densities. So, I do think that is not so called "finding" of this paper. 5. L91: "The collected core diameter was about 15 cm." I do think that the core diameter is not 15 cm according to the Geological drilling specification. Please check and correct. 6. L185, L189: the expression of "$\delta 13C‰^{'}$ is right?

---

## Referee Comment (RC3) · Anonymous Referee #3 · 30 May 2016

Unfortunately this manuscript falls short of delivering what is in the title. The authors present a very comprehensive and valuable dataset from deep boreholes. This data certainly warrants publication, but it also warrants more careful scientific analysis and context. The strength of the data is the deep boreholes, but the authors have failed to address how long term accumulation may affect SOM.

As the authors themselves point out, high-latitude regions are highly dynamic and sensitive to environmental change. Therefore the basic hypothesis that SOM at 20 m depth is controlled by the present day surface vegetation community seems rather implausible and needs further justification.

The boreholes used in this study were drilled into thick alluvial and colluvial deposits

in areas of accumulation. It is highly likely that various geomorphic process have affected the erosion as well as deposition/sedimentation of sediment in the uphill areas of these locations, together with vegetation dynamics, surface hydrology, active layer dynamics etc. has exerted a strong control over SOM distribution and chemistry. These processes have been acting over many millennia and to state that present day surface vegetation controls the SOM distribution to depths of 20 m is a gross oversimplification.

The authors should consider addressing the issue of different landforms/depositional environments instead. Table 1 provides an idea of the type of geomorphic characterization the authors can pursue to analyze these issues. The authors also show interesting analyses of the influence of soil texture on SOM and briefly mention the possibility of changes in vegetation communities over time affecting the stable isotope ratio of carbon. Pertinent follow up questions are: How is soils texture linked to landforms? What is the age of different investigated strata? Is there any link between vegetation and soil texture and/or slope stability?

I would recommend that the authors pursue these results in more depth. While vegetation seems like a useful proxy, especially since it is easy to map and scale, the authors present no evidence to support that the vegetation has remained the same in these sites over the long times when these sediments accumulated.

The statistical analyses performed show that the basic SOM chemistry follows patterns described by many other authors. They do not, however, yield any new insights into the controls of SOM in the north Qinghai-Tibet Plateau. At a more detailed level, I am also concerned that some of the unexpectedly high C:N values may be due to very low N% values possibly close to detection limit? In some cases I also wonder about the rationale behind analyses? There is no deeper mechanistic explanation provided for the correlation of % water content to SOC. Also, I would strongly recommend that the data be made available together with the final publication of this data.

With the present analyses I would not recommend that this paper is published in

[Figure]

The Cryosphere. The statistical analyses are limited and seem more spurious than hypothesis-driven. While an extensive dataset is available, I find that the authors provide little conclusions or results to significantly increase understanding of SOM accumulation or development in the north Qinghai-Tibet Plateau. I recommend that the authors take the opportunity to re-examine their extensive dataset and look critically at which landscape processes have led to the interesting SOM dynamics we see in these boreholes.

---

## Author Comment (AC1) · 11 Jul 2016

*1. Manuscript General comments*

*This manuscript is about the controls of SOM distribution in a mountain permafrost region of the QZP. The authors over simplified the factors controlling the carbon stores/density and distribution of SOC in deep strata in a permafrost environment. The discussion and conclusion (as in lines 21-23 and lines 160-169). The conclusions may well apply to the active layer or 0-2 m as described in many other papers. The near surface SOM is biogenic, resulting from the biomass accumulation form the vegetation community. However, the SOC accumulated in the deep strata may, and often the result of the geomorphic processes such as erosion and sedimentation. Therefore, the SOC stores in deep layers may not be controlled by vegetation as what is currently on the surface. Since the parent materials of these soils or cores studied are of Quaternary age, the past climate, vegetation, and especially the mode of deposition are important to the SOC stores; as we consider the syngenetic nature of the soil development.*

**Response:** Thanks for your comments. All the comments and reviews were explained as below:

(1) The effects on SOC in the active layer or upper 0-2m layer have been described in many other papers. We also hypothesized that the deep SOC may be affected by many other factors. However, these factors, such as erosion and sedimentation, were difficult to describe quantitatively and thus could not be performed a statistical analyses. In this study, we investigated the SOC in deep soils and found that the SOC was also "affected" by vegetation and

soil textures. However, as you pointed out, this opinion should not be the mechanisms of SOC accumulation. Instead, this could be a result that vegetation types and soil textures were also the result of pedogenesis and geomorphology. Thus, we added a schematic diagram and corresponding discussion to describe the relationship in the revised version (Figure 7).

Since the vegetation and soil texture data were easier available than sedimentation progress, and also could be upscaled to a regional scale, this study would be of interest to the future study in mountain permafrost regions. Therefore, in the revised version, we changed the "controls" into "close relationships" in the Conclusions and Title.

(2) To promote the conclusion in Lines 21-23:

The original version (Lines 14-17) was revised as: "To examine the pools and properties of deep SOC and their possible relationships to environmental factors, ten boreholes to the depth of about 20 m depth were drilled under alpine swamp meadow (ASM), alpine meadow (AM) and alpine steppe (AS) in the permafrost regions over Heihe River basin, Qilian Mountains."

We changed the original version (L17) "The results showed that…." was changed into "The results from these deep boreholes showed that……". We hope this revision would be helpful to potential readers for the contribution of this study.

The L21-22 was changed into "Meanwhile, the C/N ratios and carbon isotopes suggested that the SOC, which extent to tens of meters, accompanied with fine-fractions soils under swamp meadow are more decomposable than those of coarse soils at all depths. The results suggest that both the SOC stocks and chemical nature of organic matter in the upper and deep soils have close relationships with soil texture and vegetation types, which could be explained as a complex effect of geomorphology and pedogenesis."

(3) For L160-169, the original version was revised as:

"The SOC densities at AM sites (PT4, PT5, PT6, PT7) were higher than those at AS sites (PT10, PT11 and PT12), while lower than those at ASM sites (PT9, EB1, and EB2). For the upper ~3 m layers, it has been well known that vegetation types affect the SOC contents (Jobbágy and Jackson, 2000; Wu et al., 2012). This study confirmed that this pattern was not only limited to the upper layers (which were usually studied in previous reports) but also extended to deeper permafrost layers, which could reach to 5 meters (PT6, PT9, EB1, EB2) and even about 20 m depth (PT4, PT5, PT7). This study shows that the vegetation types have closely relationships with SOC distribution in deep layers in mountain permafrost areas on the north QTP."

(4) For the schematic diagram and the relationship between geomorphology, paleoclimatic conditions and SOC, we added the figure 7 in the revised version.

[Figure]

Figure 7 A schematic diagram for the relationship between environmental factors and soil organic carbon (SOC) in mountain permafrost area. The solid lines show the components of environmental conditions, arrows

show the direct effect of one factor on the other. There is also another possible effect of soil water content on the soil organic carbon via affecting the microbial growth and oxygen availability (Mu et al., 2016).

Meanwhile, the corresponding discussion was shown as follows:

"From the basic theory of SOC in permafrost carbon and results from this study, a conceptual framework was proposed as Figure 7. Topography has been long recognized as an important factor in the distribution of permafrost and soil water content (Noetzli et al., 2007), and consequently has important effects on the vegetation types (Wang et al., 2006). The landform determined sediment processes and even soil textures during pedogenesis (Yoo and Mudd, 2008). In this study, the PT9, EB1 and EB2 sites have north facing aspects with poor drainage conditions, and thus belong to swamp meadow types. The distribution of PT sites follows a pattern from mountain hills to mountain foot along with elevation gradients: (PT9, PT6) > PT7 > PT4 > PT5 > (PT10, PT11, PT12). It could be seen that drainage conditions, which usually were greatly affected by microrelief conditions (Schoeneberger, 2002), are extremely important to vegetation types (Tab 1). In QTP, previous studies showed that soil texture, vegetation, and soil water content are of great importance for the existence of permafrost (Wang et al., 2012; Wu et al., 2015). This framework was consistent with the basic theory of SOC accumulation and preservation (Genxu et al., 2012; Wu et al., 2015). It has been also known that the fine particles can protect the SOM from decomposition by the adsorption effects (Jardine et al., 1989), and soil water could be a controlling factor in microbial decomposition through limit the microbial growth and oxygen availability (Mu et al., 2016). In addition, soil water content interacts with texture and vegetation (Mohanty and Skaggs, 2001). This study showed close relationships between soil texture, water content, vegetation and SOC. Therefore, the effects of these factors

on the SOC could be both direct and indirect, which via the permafrost (Fig. 7). From this schematic diagram, it is obvious that geomorphology is the fundamental factors in the determination of SOC by the mechanisms of pedogenesis."

"The QTP is a young plateau that was uplifted since Palaeogene epoch, and the parent materials for soils distributed in the vast areas on the plateau were mainly alluvium associated mountain processes (Zheng and Yao, 2004). Therefore, the sampling area could be potentially considered as an example for the study of SOC distribution in the other areas on the QTP. Since the sampling area for PT sites is less than 100 km$^2$, and has similar meteorological conditions, thus the great differences for SOC among these sites could be attributed to the difference of topography, which affects the SOC via the pedogenesis (Fig.7). For the deep SOC stocks, the paleoclimatic conditions may also play important roles during the SOC accumulation (Schuur et al., 2009). However, this data is largely unavailable, which limited the further study of deep SOC in mountain permafrost. This study showed that the SOC both in upper layers and deep layers, which could be down to tens of meters, has close relationship with vegetation and soil texture. Although the accumulation process of SOC is difficult to be interpreted in this study due to the lack of chronological sequences of the soil layers, the results demonstrated that vegetation types and soil textures are useful proxies for the predictions of SOC in both upper and deep layers. Since these data are more accessible in regional scale (Li et al., 2015; Wang et al., 2016), it would be possible to upscale the SOC pools in the regional scale using vegetation types and soil texture data in the future."

We hope these revisions are helpful to the potential readers to understand the importance of this work and can get a clear framework of the SOC in deep permafrost-affected soils.

*2. The major problem for me to review this manuscript is the lack of original data. There is no tabulated data for each analyzed soil horizon or layer. Thus there is no way to tell if the values of pH, soil texture, conductivity, C-density, C/N ratio presented are from one particular section/horizon or the average of the whole sampling depth. The authors are responsible should provide the original data as supplement that should include all the analyses as indicated in the Method section, and present the analytical data of each sampled layers of sections.*

**Response:** Sorry for the confusion. We are pleased provide the original data during the submission. We also submitted it as supplementary file. Unfortunately, it could not be found in the TCD. Therefore, in the revised version, we explained it clearly and gave the link of the data, thus potential authors can realize we also submit this regional data.

In the revised version, this had been explained as:

"All the original data were available on the website of The Cryosphere Discussions (http://editor.copernicus.org/index.php?_mdl=msover_md&_jrl=25&_lcm=oc73lcm74a&_acm= get_supplement_file&_ms=50278&id=704538&salt=1523858444357408567)."

*3. The soils are likely syngenetic if the particle size distribution is more relatively uniform, or fluvial/erosion or sedimentation modification if there are contrasting soil textures. The %SOC correlates well with %clay because of physical protection, a function of surface area. But the fine soil particles are not limited to clay. There are several papers dealing with SOC contents in both the clay and silt fractions. Will there be any difference if the %silt is considered in the correlation?*

**Response:** Thanks for the review. From the parent materials, the soils are likely syngenetic. We also hoped that there would be a statistically significant correlation between the silt content and

SOC, but the silt was excluded from the stepwise linear regression, and the *Pearson* correlation was not significant. The non-significant relationship between the silt content and SOC could be due to the methods for size distribution, because the laser diffraction instrument only showed the proportions of silt, while without the detailed information about very fine silt, fine silt, medium silt and coarse silt.  This was discussed in the revised version as below:

"In this study, the clay content was significantly correlated with the SOC content. This finding is in agreement with the reports for the upper 2 m layers (Wu et al., 2015) and can be explained by the presence of fine particles, which tend to stabilize and retain more organic matter than coarser particles (Gregorich et al., 1994). In addition, fine particles have a higher water holding capacity (Gómez-Plaza et al., 2001). The fine particles are not confined with clay, other proportions such as very fine silt, fine silt, may also relate to the SOC contents (Vogel et al., 2015). However, this study showed no significant relationship between SOC and silt content. The non-significant relationship between the silt content and SOC could also be due to the methods for size distribution, because the laser diffraction instrument only showed the proportions of silt, while without detailed information about very fine silt, fine silt, medium silt and coarse silt."

*4. Soil drainage is mentioned in Table 1 but not discussed. The SOC content is controlled by vegetation community which is affected by drainage or soil water content due to soil texture and landform position. I recommend the manuscript be accepted upon major revision.*

**Response:** Thanks for the suggestion. It is true that soil water content was affected by soil texture, landform position and aspect, meanwhile, soil water influences the distribution of

vegetation community, which can determine SOC content. It is a closely and inseparable relationship.

In the revised version, this was discussed via the schematic as below:

"In this study, the PT9, EB1 and EB2 sites have north facing aspects with poor drainage conditions, and thus belong to swamp meadow types. The distribution of PT sites follows a pattern from mountain hills to mountain foot along with elevation gradients: (PT9, PT6) > PT7 > PT4 > PT5 > (PT10, PT11, PT12). It could be seen that drainage conditions, which usually were greatly affected by microrelief conditions (Schoeneberger, 2002), are extremely important to vegetation types (Tab 1)."

The soil water content was also discussed (Response to Question 1).

*5. Specific comments*

*L. 18. "silt loam over ASM" change "over" to "in". L. 19-20. "higher fine-fractions" change to "higher finer textured fractions"; "coarse soils" change to "coarser-textured soils".*

**Response:** Thanks. Changed.

*6. L. 21. "more decomposable" or "more decomposed"?*

**Response:** Thanks. It should be "more decomposed".

*7. L. 29. Insert "more" after "become". Insert "due to climate warming" after "decomposition".*

**Response:** Thanks. Changed.

*8. L. 34. Delete "permafrost".*

**Response:** Deleted.

*9. L. 38. Citation "(Bockheim and Hinkel, 2007)" is not the proper reference. In their paper, the deepest soil horizon was sampled at 161 cm. This can hardly be called "deep carbon". For the vulnerability of deep carbon, refer to the papers by Schuur et al. and Zimov et al.*

**Response:** Thanks very much. The references were changed.

*10. L. 46-48. "consequently resulting in easily decomposable substances". It is because the original substrate was easily decomposable thus resulting in less negative delta $^{12}C$ values. So consequently resulting in highly decomposed substances as indicated by lower C/N values.*

**Response:** Thanks. It was changed as below:

"$^{12}C$ is preferentially used by decomposers, which may lead to $^{13}C$ enrichment in the remaining SOC (Natelhoffer and Fry, 1988), because the original substrate was easily decomposable thus resulting in less negative delta $\delta^{12}C$ values, consequently resulting in highly decomposed substances as indicated by lower C/N values. Therefore, the C/N ratios and $\delta^{13}C$ values of organic matter can be used to reveal the potential for microbial decomposition of organic matter in permafrost regions."

*11. L. 60. "soil texture also relates to vegetation types". Soil texture is one of the several factors affecting vegetation types.*

**Response:** Thanks. Changed into "Moreover, soil texture is one of the several factors affecting vegetation types, soil moisture and even the thickness of the active layer of permafrost"

*12. L.64. Insert "carbon" between "the" and "contents".*

**Response:** Thanks, inserted. The first hypothesis was changed into "(1) the distribution of SOC in the soils, including surface and deep layers, has close relationships to vegetation types and soil texture on the QTP; (2) there are significant differences in the characteristics of SOC under different vegetation types and different soil texture classes."

*13. L. 70. Change "area" to "region".Last word "westerlies"?*

**Response:** Thanks. Changed into "region"

It should be "Prevailing Westerlies". Because it has nothing to do with the topic, we deleted this in the revised version.

*14. L. 71. Insert "mean" before "annual".*

**Response:** Thanks. Inserted.

*15. L. 74. Change "main"to "major".*

**Response:** Thanks. Changed.

*16. L. 76. Capitalize "Quaternary"/*

**Response:** Thanks. Capitalized.

*17. L. 80. Change "gradually" to "gradual".*

**Response:** Thanks. Changed.

*18. L. 86. Last word "largely" change to "strongly". Are the pH values and electric conductivity measured for only the surface soil (topsoils) or average values for the whole profile (down to the bottom of sampling? For definition for saline and alkaline soils, see http://www.nrcs.usda.gov/Internet/FSE_DOCUMENTS/nrcs142p2_052523.pdf Reaction of EB1 is neutral and EB2 is slightly alkaline.*

**Response:** Thanks. The "largely" was changed into "strongly". The pH and EC values were for the active layers, and this was clarified in the revised version. Thanks for the references, and the soils for EB1 and EB2 were corrected into neutral and slightly alkaline in the revised version. This sentence was revised as below:

"The soils in the study area were strongly alkaline, with pH values of 8.53-8.84, with the exception of EB1 (neutral) and EB2 (slightly alkaline). The conductivities of the soil suspensions ranged from 0.95 to 1.33 ms cm$^{-1}$ (Table 2, data for the active layers, and the upper 2 m for the PT11)."

*19. L. 99. Delete "values" and change "conductivities" to singular.*

**Response:** Thanks, changed into "pH and conductivity".

*20. L. 100-103. Where is the data for soil texture and rock fragment content?*

**Response:** Thanks for the review. For the soil texture analysis, we firstly removed the gravels (rocks) and calculated the ratios by weight methods, then these samples were used for laser diffraction instrument. In the revised version, we changed the rock into gravel. These data were presented as supplementary materials in the revised version as below:

"All the original data were available on the website of The Cryosphere Discussions (http://editor.copernicus.org/index.php?_mdl=msover_md&_jrl=25&_lcm=oc73lcm74a&_acm= get_supplement_file&_ms=50278&id=704538&salt=1523858444357408567)."

*21. L. 103. Insert "contents" after "(TN)".*

**Response:** Inserted.

*22. L. 120. The variation of vegetation type is limited to 2 Kobresia species.*

**Response:** Thanks, the "different" was removed in the revised version.

*23. L. 122. The SOC density ranged from 0.4 to 22.4 kg m$^{-3}$. Is the density of different soil horizon or this is the average of the whole soil profile? If so, then what is the carbon stores (kg m$^{-2}$) of the active layer or 0-2 m and the whole profile?*

**Response:** Thanks. This is for the soil layers at different depths. In this paper, the main goal is to discuss the relationship between carbon densities and the textures, especially for the deep layers. Since the SOC stores for the 0-2 m and the whole profile are of interesting for potential readers, we calculated these values in the results section.

"As shown in Table 2, the SOC stocks for the upper 2 m were highest for ASM sites (varied from 38.39 to 58.20 kg m$^{-2}$), followed by AM sites (varied from 8.62 to 21.73 kg m$^{-2}$). The lowest values appeared in AS sites (lower than 5.0 kg m$^{-2}$). For all the sites, the most SOC was distributed in the upper 6 m. The upper 6 m SOC stocks showed similar trends with those of upper 2 m. The highest SOC was recorded at EB1 site, while the PT9 had higher SOC stocks

than that of EB2 since the later had a shallower soil thickness. The SOC stocks for the upper 6 m

layers at AM sites varied from 29.7 to 48.5 kg m$^{-2}$. The SOC stocks were lowest at AS sites."

Table 2 SOC stocks (SOCC, kg m$^{-2}$) for different layers for the sampling sites

| Site | 0-1 m | 0-2 m | 0-3 m | 0-6 m | Active layer |
|---|---|---|---|---|---|
| PT4 | 9.74±0.62 | 10.81±1.35 | 18.17±1.67 | 38.04±2.09 | 3.63±0.44 |
| PT5 | 8.94±0.65 | 16.05±1.21 | 20.37±1.87 | 29.72±3.01 | 3.42±0.38 |
| PT6 | 11.84±0.88 | 21.73±2.04 | 29.47±3.08 | 48.51±4.33 | 2.40±0.14 |
| PT7 | 5.20±0.48 | 8.62±0.75 | 13.20±1.43 | 29.89±3.05 | 2.41±0.17 |
| PT9 | 22.76±2.14 | 38.39±3.66 | 57.46±6.35 | 104.17±7.76 | 1.63±0.09 |
| EB1 | 39.62±3.17 | 58.20±4.43 | 81.88±7.77 | 134.46±9.94 | 1.20±0.05 |
| EB2 | 34.49±2.43 | 52.89±3.20 | 64.24±4.31 | 69.47±5.66 | 1.30±0.04 |
| PT10 | 3.85±0.11 | 3.91±0.18 | 4.07±0.32 | 4.66±0.38 | 4.85±0.31 |
| PT11 | 3.91±0.22 | 4.70±0.27 | 5.24±0.37 | 7.25±0.67 | 6.00±0.60 |
| PT12 | 0.55±0.04 | 1.10±0.08 | 2.36±0.14 | 7.59±0.51 | 5.75±0.43 |

Data were presented as Mean±SD from measurements of three triplicate samples.

*23. L. 128. Add "respectively" at the end of sentence.*

**Response:** Thanks, added.

*24. L. 141. Insert "class" after "texture".*

**Response:** Inserted.

*25. L. 151. "moisture" is not the proper word; water! The equation is poorly constructed. Use symbols; in line 150, add (D) after depth, add (W) after water content, add (Db) after bulk density, and "Cy" or other choice after clay content.*

**Response:** Thanks, it was changed as below:

According to the relationship between the SOC contents and depth (D), total water content (W), Gravel (G), and clay content (Cy), the best regression models were as follows:

SOC(%)=0.092W-0.214D-0.20lG+0.672Cy; $r^2$=0.566, p<0.001 (n=158).

*26. L. 155. See general comments.*

**Response:** Thanks, see the responses to Question 1-4.

*27. Tables Need footnote for the Drainage class in Table 1. Why are there 2 columns of conductivity? Tables 1 and 2 should be combined and titled "Physiographic environment of the study sites in the Heihe River basin, Qinghai_Tibetan Plateau". Soil properties should be in another table. Besides pH, conductivity, the methods section also include SOC and water contents, C/N ratio, bulk density, particle size distribution, rock fragment content.*

**Response:** Sorry for the mistakes for the conductivity. The tables were reformed and revised according to your suggestions. The combined table was presented at Table 1 as below:

For the soil properties, the table is too large, and could not be presented in the paper, therefore, we submitted it as supplementary file. See response to Question 20.

Table 1 Physiographic environment of the study sites over Heihe River basin, Qilian Mountains

| Site | Longitude (°) | Latitude (°) | Altitude (m) | ALT (m) | Aspect | Slope (°) | Topography | Drainage class | BD (m) | SLT (m) | Vegetation types | Vegetation cover% | Dominant species | pH | Conductivity (ms cm$^{-1}$) |
|------|-----------|----------|----------|------|--------|-------|-----------|-----------|-----|------|------------|------------|-----------|------|--------------|
| PT4 | 98.946 | 38.833 | 3770 | 3.63 | southeast | flat | PP | P | 90 | 32.5 | AM | 94 | *Kobresia pygmaea* | 8.53 | 1.11 |
| PT5 | 99.026 | 38.806 | 3692 | 3.42 | southeast | flat | PP | P | 20 | >20* | AM | 90 | C. B. Clarke | 8.84 | 0.95 |
| PT6 | 98.963 | 38.955 | 4159 | 2.40 | southeast | 2 | PS | P | 50 | 9 | AM | 80 | *Ajania tibetica* | 8.64 | 1.05 |
| PT7 | 98.963 | 38.903 | 3970 | 2.41 | southeast | 1.5 | PP | P | 36 | 25.5 | AM | 85 | *Rhodiola subopposita* | 8.70 | 1.09 |
| PT9 | 98.950 | 38.627 | 4138 | 1.63 | northeast | 2 | PS | VP | 160 | 7 | ASM | 96 | *K.tibetica Maxim* | 8.65 | 1.12 |
| PT10 | 99.068 | 38.789 | 3681 | 4.85 | flat | flat | PP | SE | 20 | >20 | AS | 87 | | 8.76 | 1.14 |
| PT11 | 99.068 | 38.789 | 3680 | SFG | flat | flat | PP | SE | 20 | >20 | AS | 78 | *K. humilis (C.A.Mey.) Serg* | 8.61 | 1.03 |
| PT12 | 99.068 | 38.788 | 3680 | 5.75 | flat | flat | PP | SE | 20 | >20 | AS | 75 | | 8.80 | 1.83 |
| EB1 | 100.916 | 37.998 | 3700 | 1.2 | northwest | 1.2 | PS | VP | 20 | 8** | ASM | 95 | *K.tibetica Maxim* | 6.58 | 1.02 |
| EB2 | 100.907 | 38.003 | 3615 | 1.3 | northwest | 2.5 | PS | VP | 11.7 | 8*** | ASM | 95 | *K.tibetica Maxim* | 7.44 | 1.33 |

ALT: Active layer thickness; BD: Borehole depth; SLT: Soil layer thickness; MAGT: Mean annual ground temperature; ASM: Alpine swamp meadow; AM: Alpine meadow; AS: Alpine steppe. SFG: Seasonal frozen ground.

PP: Piedmont plain; PS: Piedmont slope;

*Clay for the soils below; 16 m; ** High ice contents for 6-8 m; *** high gravel contents for 5-8 m

---

## Author Comment (AC2) · 11 Jul 2016

*1. This manuscript tried to clarify the main factors for affecting the SOC densities in the Tibetan Plateau based on dataset from the far north corner of the Tibetan Plateau, source region of Heihe River, Qilian Mountains. The dataset was valuable in such a data-absent region, specially, the dataset from deeper layer of soils. But the conclusions were general knowledge, which could be found in most related literatures.*

**Response:** Thanks for your acknowledgment of the merits of the data. The general knowledge usually confined to the upper soil layers and it seldom involved the SOC with depth deeper than 2 or 3 m. Moreover, little is concerned about the effects of vegetation and soil texture on deep carbon in mountain permafrost regions.

Since the deep frozen carbon pools in permafrost regions are important to global carbon budget under global warming scenarios, this finding would be helpful to improve our knowledge in the carbon pools in mountain permafrost areas, where with great heterogeneities. In the revised version, we clarified these points. Detailed information was included in the responses of questions 2-7.

*2. The concerns and suggestions include:*

*It could be seen from Figure 1 that all the boreholes are located on the bottom of valley and the lower part (gentle) of the slopes. The sites even could not cover all the surface ecological and geological conditions of the study region. The dataset is so limited, just 10 boreholes to be considered as representatives of 3 types of ground surface conditions (AS, AM and ASM). Furthermore, the sampling sites are located in the far northeast boundary of the Plateau, and the area of the study region accounts for less than ten thousandth of Tibetan Plateau. All the geologic, geomorphologic, geographical and climatic backgrounds are great different from the real plateau. I do think that the dataset collected in this region just can be representative of the local condition, even not as representative of Qilian Mountain Ranges, because the climatic conditions is also great different to the western part of the mountain range. It would be better if the title of the manuscripts revised as "Controls on the distribution of the soil organic matter in the Upper Reach of Heihe River, Qilian Mountains.*

**Response:** Thanks very much for your suggestions. We changed the tile into "Close relationships between deep organic carbon and soil texture with vegetation types in permafrost regions over Heihe River basin, Qilian Mountains, China".

We proposed a schematic diagram (Figure 7) in the revised version, and we explained in the discussion that the result potentially is applicable for the other areas of Qinghai-Tibetan Plateau as below:

"The QTP is a young plateau that was uplifted since Palaeogene epoch, and the parent materials for soils distributed in the vast areas on the plateau were mainly alluvium that associated mountain processes (Zheng and Yao, 2004). Therefore, the sampling area could be

potentially considered as an example for the study of SOC distribution for the other areas on the QTP. Since the sampling area for PT sites is less than 100 km$^2$, and has same meteorological conditions, thus the great differences for SOC among these sites could be attributed to the difference of topography, which affects the SOC via the pedogenesis (Fig. 7). For the deep SOC stocks, the paleoclimatic conditions may also played important roles during the SOC accumulation (Schuur et al., 2009). However, this data is largely unavailable, which limited the further study of deep SOC in mountain permafrost. This study showed that the SOC both in upper layers and deep layers, which could be down to tens of meters, has close relationship with vegetation and soil texture. Although the accumulation process of SOC is difficult to be interpreted in this study due to the lack of chronological sequences of the soil layers, the results demonstrated that vegetation types and soil textures are useful proxies for the predictions of SOC in both upper and deep layers. Since these data are more accessible in regional scale (Li et al., 2015; Wang et al., 2016), it would be possible to upscale the SOC pools in the regional scale using vegetation types and soil texture data in the future."

*3. The results in section 3.1 and 3.2 are very general description for soil organic carbon, C:N ratios and stable carbon isotopes. The highest soil organic carbon density was found in boreholes under ASM, and the lowest was at AS. Similar results were reported in great amount of literatures by Wang, et al. and Wu et al., but there is no more new.*

**Response:** Thanks very much. The previous study confined the SOC contents and C:N ratios to the upper layers, however, little is known about their distribution in deep soils. Therefore,

we performed this study. In the revised version, we clarified this and emphasized the SOC in deep layers.

In the section 3.1, we emphasized as: "From the upper layers to the deep depth, which down to 20 m, ASM sites (PT9, EB1 and EB2), the SOC densities were much higher than those of AM sites, although there was a decreasing trend along with depth at EB2. The mean SOC densities for the sites ranged from 0.4 to 22.4 kg m$^{-3}$ at different depth…."

We also added the SOC stocks for the different depths, this result may be of interest potential readers since carbon stocks below 2 m depth are rare.

"As shown in Table 2, the SOC stocks for the upper 2 m were highest for ASM sites (varied from 38.39 to 58.20 kg m$^{-2}$), followed by AM sites (varied from 8.62 to 21.73 kg m$^{-2}$). The lowest values appeared in AS sites (lower than 5.0 kg m$^{-2}$). For all the sites, the most SOC was distributed in the upper 6 m. The upper 6 m SOC stocks showed similar trends with those of upper 2 m. The highest SOC was recorded at EB1 site, while the PT9 had higher SOC stocks than that of EB2 since the later had a shallower soil thickness. The SOC stocks for the upper 6 m layers at AM sites varied from 29.7 to 48.5 kg m$^{-2}$. The SOC stocks were lowest at AS sites."

Table 2 SOC stocks (SOCC, kg m$^{-2}$) for different layers for the sampling sites

| Site | 0-1 m | 0-2 m | 0-3 m | 0-6 m | Active layer |
|------|-------|-------|-------|-------|--------------|
| PT4 | 9.74±0.62 | 10.81±1.35 | 18.17±1.67 | 38.04±2.09 | 3.63±0.44 |
| PT5 | 8.94±0.65 | 16.05±1.21 | 20.37±1.87 | 29.72±3.01 | 3.42±0.38 |
| PT6 | 11.84±0.88 | 21.73±2.04 | 29.47±3.08 | 48.51±4.33 | 2.40±0.14 |
| PT7 | 5.20±0.48 | 8.62±0.75 | 13.20±1.43 | 29.89±3.05 | 2.41±0.17 |
| PT9 | 22.76±2.14 | 38.39±3.66 | 57.46±6.35 | 104.17±7.76 | 1.63±0.09 |
| EB1 | 39.62±3.17 | 58.20±4.43 | 81.88±7.77 | 134.46±9.94 | 1.20±0.05 |
| EB2 | 34.49±2.43 | 52.89±3.20 | 64.24±4.31 | 69.47±5.66 | 1.30±0.04 |
| PT10 | 3.85±0.11 | 3.91±0.18 | 4.07±0.32 | 4.66±0.38 | 4.85±0.31 |
| PT11 | 3.91±0.22 | 4.70±0.27 | 5.24±0.37 | 7.25±0.67 | 6.00±0.60 |
| PT12 | 0.55±0.04 | 1.10±0.08 | 2.36±0.14 | 7.59±0.51 | 5.75±0.43 |

Data were presented as Mean ±SD from measurements of three triplicate samples.

For the 3.2 section, we emphasized as below:

"For samples at different depths including below 2 m depth, the C/N ratio and SOC content had a weak positive relationship for ASM sites ($r^2$=0.028, p<0.05, Fig. 3a), whereas they had a higher correlation for AS and AM sites ($r^2$=0.522, p<0.001, Fig. 3b)….."

We hope these revisions would be helpful to highlight the merits of the data for the deep SOC in the permafrost regions.

*4. SOC in deeper soil layers should be affected more by paleo-climatic, ecological and geological background of the soil formation. The authors simply correlated SOC with the moisture content and texture (gravel and clay) of the soils. It would be better to add more*

*information about soil formation history and discuss the controlling factors of SOC for different soil layers separately.*

**Response:** Thanks for the review. The paleo-climatic, ecological and geological background of the soil formation should be the fundamental mechanisms for the SOC formation and preservation in deep layers. We tried to add such information in the data analysis, however, these data were either largely unavailable or could not be presented quantitatively for statistically analysis. Therefore, we changed the title into "Close relationships between deep organic carbon and soil texture with vegetation types in permafrost regions over Heihe River basin, Qilian Mountains, China"

According to your suggestions, we discussed these in the revised version as below:

"From the basic theory of SOC in permafrost carbon and results from this study, a conceptual framework was proposed as Figure 7. Topography has been long recognized as an important factor in the distribution of permafrost and soil water content (Noetzli et al., 2007), and consequently has important effects on the vegetation types (Wang et al., 2006). The landform determined sediment processes and even soil textures during pedogenesis (Yoo and Mudd, 2008). In this study, the PT9, EB1 and EB2 sites have north facing aspects with poor drainage conditions, and thus belong to swamp meadow types. The distribution of PT sites follows a pattern from mountain hills to mountain foot along with elevation gradients: (PT9, PT6) > PT7 > PT4 > PT5 > (PT10, PT11, PT12). It could be seen that drainage conditions, which usually were greatly affected by microrelief conditions (Schoeneberger, 2002), are extremely important to vegetation types (Tab 1). In QTP, previous studies showed that soil texture, vegetation, and soil water content are of great importance for the existence of

permafrost (Wang et al., 2012; Wu et al., 2015). This framework was consistent with the basic theory of SOC accumulation and preservation (Wang et al., 2012; Wu et al., 2015). It has been also known that the fine particles can protect the SOM from decomposition by the adsorption effects (Jardine et al., 1989), and soil water could be a controlling factor in microbial decomposition through limit the microbial growth and oxygen availability (Mu et al., 2016). In addition, soil water content interacts with texture and vegetation (Mohanty and Skaggs, 2001). This study showed close relationships between soil texture, water content, vegetation and SOC. Therefore, the effects of these factors on the SOC could be both direct and indirect, which via the permafrost (Fig. 7). From this schematic diagram, it is obvious that geomorphology is the fundamental factors in the determination of SOC by the mechanisms of pedogenesis."

[Figure]

Figure 7 A schematic diagram for the relationship between environmental factors and soil organic carbon (SOC) in mountain permafrost area. The solid lines show the components of environmental conditions, arrows show the direct effect of one factor on the other. There is also another possible effect of

soil water content on the soil organic carbon via affecting the microbial growth and oxygen availability

(Mu et al., 2016).

"The QTP is a young plateau that was uplifted since Palaeogene epoch, and the parent materials for soils distributed in the vast areas on the plateau were mainly alluvium associated mountain processes (Zheng and Yao, 2004). Therefore, the sampling area could be potentially considered as example for the study of SOC distribution for the other areas on the QTP. Since the sampling area for PT sites is less than 100 $km^2$, and has similar meteorological conditions, thus the great differences for SOC among these sites could be attributed to the difference of topography, which affects the SOC via the pedogenesis (Fig.7). For the deep SOC stocks, the paleoclimatic conditions may also played important roles during the SOC accumulation (Schuur et al., 2009). However, this data is largely unavailable, which limited the further study of deep SOC in mountain permafrost. This study showed that the SOC both in upper layers and deep layers, which could be down to tens of meters, has close relationship with vegetation and soil texture. Although the accumulation process of SOC is difficult to be interpreted in this study due to the lack of chronological sequences of the soil layers, the results demonstrated that vegetation types and soil textures are useful proxies for the predictions of SOC in both upper and deep layers. Since these data are more accessible in regional scale (Li et al., 2015; Wang et al., 2016), it would be possible to upscale the SOC pools in the regional scale using vegetation types and soil texture data in the future."

We hope these revisions are helpful to the potential readers to get a clear framework of the SOC in deep soils.

*5. L160-168: it is a general knowledge that SOC is produced by photosynthesis of plants. There is without exception for organic carbon deposited in deep soil layers. Therefore, generally speaking, the better in the vegetation, and the higher in SOC densities. So, I do think that is not so called "finding" of this paper.*

**Response:** Thanks. We revised the text as the follows:

   "For the upper ~3 m layers, it has been well known that vegetation types affect the SOC contents (Jobbágy and Jackson, 2000; Wu et al., 2012). Result from this study confirmed that this pattern was not only limited to the upper layers (which were usually studied in previous reports) but also extended to the deep permafrost layers, which could reach to 5 meters (PT6, PT9, EB1, EB2) and even about 20 m depth (PT4, PT5, PT7)."

   We hope these sentences would be helpful to the potential readers so that they can understand the main propose of this study.

*6. L91: "The collected core diameter was about 15 cm." I do think that the core diameter is not 15 cm according to the Geological drilling specification. Please check and correct.*

**Response:** Thanks for your reminding. The diameter was 13.5 cm for the upper 20 cm and then changed to 11.7 below 20 m. Since this data only collected from upper 20 m layers, we clarified it as "13.5 cm".

*7. L185, L189: the expression of "$_{}^{13}C$‰'' is right?*

**Response:** Thanks, we deleted the "‰" in the revised version.

---

## Author Comment (AC3) · 11 Jul 2016

1. Unfortunately, this manuscript falls short of delivering what is in the title. The authors present a very comprehensive and valuable dataset from deep boreholes. This data certainly warrants publication, but it also warrants more careful scientific analysis and context. The strength of the data is the deep boreholes, but the authors have failed to address how long term accumulation may affect SOM.

**Response:** Thank you very much for your helpful review. We appreciate your acknowledgement of the difficulty in obtaining this data from deep boreholes. We revised this manuscript in the revised version according your comments and suggestion. Detailed information was listed as the responses.

2. As the authors themselves point out, high-latitude regions are highly dynamic and sensitive to environmental change. Therefore, the basic hypothesis that SOM at 20 m depth is controlled by the present day surface vegetation community seems rather implausible and needs further justification.

**Response:** Thanks. We realized that the fundamental mechanisms for the SOC accumulation and preservation were the soil formation and paleoclimatic conditions. However, these data were largely unavailable or difficult to perform statistical analysis since they are difficult to be

described quantitatively. In contrast, the vegetation and soil texture are more accessible, therefore, if close relationships between vegetation and soil texture could be found, it would be potentially helpful to upscale deep SOC to a regional scale in the future.

We expected these relationships based on three concepts: 1) In a small area, the climatic conditions are similar; 2) The difference of present vegetation reflects the topography and pedogenesis for different sites; 3) The differences of topography and pedogenesis among different sites have been lasted long time during the accumulation of the SOC.

The statistical analysis showed there were close relationships between the SOC and soil textures and vegetation types and the relationship should be explained as that these factors reflect the process of SOC accumulation and preservation instead of controlling factors. Therefore, in the revised version, we changed the title into "Close relationships between deep organic carbon and soil texture with vegetation types in permafrost regions over Heihe River basin, Qilian Mountains, China".

To clarify the relationships between SOC and vegetation and soil texture, we added the effects of soil formation process on the Qinghai-Tibetan Plateau in the discussion section in the revised version as below:

"From the basic theory of SOC in permafrost carbon and results from this study, a conceptual framework was proposed as Figure 7. Topography has been long recognized as an important factor in the distribution of permafrost and soil water content (Noetzli et al., 2007), and consequently has important effects on the vegetation types (Wang et al., 2006). The landform determined sediment processes and even soil textures during pedogenesis (Yoo and Mudd, 2008). In this study, the PT9, EB1 and EB2 sites have north facing aspects with poor drainage

conditions, and thus belong to swamp meadow types. The distribution of PT sites follows a pattern from mountain hills to mountain foot along with elevation gradients: (PT9, PT6) > PT7 > PT4 > PT5 > (PT10, PT11, PT12). From the vegetation types, it could be seen that drainage conditions, which usually were greatly affected by microrelief conditions (Schoeneberger, 2002), are extremely important to vegetation types (Tab 1). In QTP, previous studies showed that soil texture, vegetation, and soil water content are of great importance for the existence of permafrost (Wang et al., 2012; Wu et al., 2015). This framework was consistent with the basic theory of SOC accumulation and preservation (Wang et al., 2012; Wu et al., 2015). It has been also known that the fine particles can protect the SOM from decomposition by the adsorption effects (Jardine et al., 1989), and soil water could be a controlling factor in microbial decomposition through limit the microbial growth and oxygen availability (Mu et al., 2016). In addition, soil water content interacts with texture and vegetation (Mohanty and Skaggs, 2001). This study showed close relationships between soil texture, water content, vegetation and SOC. Therefore, the effects of these factors on the SOC could be both direct and indirect, which via the permafrost (Fig. 7). From this schematic diagram, it is obvious that geomorphology is the fundamental factors in the determination of SOC by the mechanisms of pedogenesis."

Figure 7 A schematic diagram for the relationship between environmental factors and soil organic carbon (SOC) in mountain permafrost area. The solid lines show the components of environmental conditions, arrows show the direct effect of one factor on the other. There is also another possible effect of soil water content on the soil organic carbon via affecting the microbial growth and oxygen availability (Mu et al., 2016).

"The QTP is a young plateau that was uplifted since Palaeogene epoch, and the parent materials for soils distributed in the vast areas on the plateau were mainly alluvium associated mountain processes (Zheng and Yao, 2004). Therefore, the sampling area could be potentially considered as example for the study of SOC distribution for the other areas on the QTP. Since the sampling area for PT sites is less than 100 km2, and has similar meteorological conditions, thus the great differences for SOC among these sites could be attributed to the difference of topography, which affects the SOC via the pedogenesis (Fig.7). For the deep SOC stocks, the paleoclimatic conditions may also played important roles during the SOC accumulation (Schuur et al., 2009). However, this data is largely unavailable, which limited the further study of deep

SOC in mountain permafrost. This study showed that the SOC both in upper layers and deep layers, which could be down to tens of meters, has close relationship with vegetation and soil texture. Although the accumulation process of SOC is difficult to be interpreted in this study due to the lack of chronological sequences of the soil layers, the results demonstrated that vegetation types and soil textures are useful proxies for the predictions of SOC in both upper and deep layers. Since these data are more accessible in regional scale (Li et al., 2015; Wang et al., 2016), it would be possible to upscale the SOC pools in the regional scale using vegetation types and soil texture data in the future."

3. The boreholes used in this study were drilled into thick alluvial and colluvial deposits in areas of accumulation. It is highly likely that various geomorphic process have affected the erosion as well as deposition/sedimentation of sediment in the uphill areas of these locations, together with vegetation dynamics, surface hydrology, active layer dynamics etc. has exerted a strong control over SOM distribution and chemistry. These processes have been acting over many millennia and to state that present day surface vegetation controls the SOM distribution to depths of 20 m is a gross oversimplification. The authors should consider addressing the issue of different landforms/depositional environments instead. Table 1 provides an idea of the type of geomorphic characterization the authors can pursue to analyze these issues.

**Response:** Thanks very much for your valuable comments. After carefully looking over the sampling sites, we realized that the soil thickness was related to the geomorphic conditions, thus the pedogenesis affected the SOC accumulation and preservation. After a carefully consideration, the geomorphic characterization is difficult to describe quantitatively and could not be performed

statistically analysis. Therefore, we presented a schematic diagram (Figure 7) in the revised version to discuss about this. The revisions were seen as the response to Question 2.

4. The authors also show interesting analyses of the influence of soil texture on SOM and briefly mention the possibility of changes in vegetation communities over time affecting the stable isotope ratio of carbon.

**Response:** Thanks for the comments.

5. Pertinent follow up questions are: How is soils texture linked to landforms? What is the age of different investigated strata? Is there any link between vegetation and soil texture and/or slope stability? I would recommend that the authors pursue these results in more depth. While vegetation seems like a useful proxy, especially since it is easy to map and scale, the authors present no evidence to support that the vegetation has remained the same in these sites over the long times when these sediments accumulated.

**Response:** Thanks very much for these interesting questions. The statistically analysis showed that close relationship between the SOC and texture and vegetation types. However, this should not be explained as the controlling factors, it should be more appropriate to attribute the accumulation of SOM to the soil formation process, which links to the factors you mentioned above. As our response to Question 3, these factors were difficult to describe quantitatively and could not perform statistical analysis, we present a schematic diagram to discuss the relationship to landform, as well as the links between the vegetation and soil texture and slope.

(1) *How is soil texture linked to landforms?*

This was revised as below:

"From the soil textures (Fig. 5), it could be found that the ASM sites were mainly silt loams (EB1, EB2, and PT9), AS sites (PT10, PT11, PT12) were mainly sandy loams, while other sites (PT4, PT5, PT6, PT7) have both silt loans and sandy loams, which depended on the depth. This pattern was closely related to the locations of these sites (Fig. 1), i.e., ASM sites located at hillslopes, AM sites located at mountain valley. Interestingly, and the soil thickness, which is also a factor affects SOC because it is an independent factor in the calculation of SOC stocks for a certain depth, largely showed an opposite trend (Fig. 2). From the modes of pedogenesis in mountain area (de Vente and Poesen, 2005; Dietrich and Dunne, 1978), it could be explained as the fine particles in mountain foot and mountain valley have been transported by water. In many areas, the stability of slope usually plays an important role in vegetation (Greenway, 1987; Norris et al., 2008). The slopes of the sampling sites in this study were always smaller than 2.5°, and even with flat landform. The sampling sites were selected at areas without signals of instability of slope during the field work. Thus the slope stability seems not to be an important factor affecting vegetation and soil textures in the present study. Overall, topography, which mainly consists of slope, aspect, and landform, greatly affects the soil texture and further affects the SOC pools."

**(2) What is the age of different investigated strata?**

This is of great interesting questions pertinent to the SOC accumulation process. Unfortunately, we did not perform the chronological analysis of these samples. Although our goal is to study the patterns of the deep SOC in permafrost regions, we clarified this in the revised version as below:

"Although the accumulation process of SOC is difficult to be interpreted in this study due to the lack of chronological sequences of the soil layers, the results demonstrated that vegetation types and soil textures are useful proxies for the predictions of SOC in both upper and deep layers."

**(3) Is there any link between vegetation and soil texture and/or slope stability?**

The relationship between vegetation and soil texture was explained as Figure 7 (See response to Question 3).

For the slope stability, we discussed as below:

"In many areas, the stability of slope usually plays an important role in vegetation (Greenway, 1987; Norris et al., 2008). The slopes of sampling sites in the study were always smaller than 2.5°, and even with flat landform. The sampling sites were selected at areas without signals of instability of slope during the field work. Thus the slope stability seems not to be an important factor affecting vegetation and soil textures in the present study. Overall, topography, which mainly consists of slope, aspect, and landform, greatly affects the soil texture and further affects the SOC pools."

We hope these work would be helpful to the potential readers so they can understand the soil formation process need future work to explain the effects on the process of SOC accumulation.

6. The statistical analyses performed show that the basic SOM chemistry follows patterns described by many other authors. They do not, however, yield any new insights into the controls of SOM in the north Qinghai-Tibet Plateau. At a more detailed level, I am also concerned that some of the unexpectedly high C:N values may be due to very low N% values possibly close to detection limit? In some cases I also wonder about the rationale behind analyses? There is no deeper mechanistic explanation provided for the correlation of % water content to SOC. Also, I

would strongly recommend that the data be made available together with the final publication of this data.

**Response:** Thanks for your interesting on our original data and encourage us to dig deeper in the data. Our main goal was to explore the patterns of the SOC in deep soils. This was emphasized in the revised version.

We have carefully checked the original data of the nitrogen, the nitrogen contents were largely higher than 0.2 g/kg, which were above the detect limit of the elemental analyzer. In addition, the samples have been analyzed in triplicate, and thus the high C:N ratios were reliable in this study.

The rationale behind the analyses could be excluded from the statistical analysis. However, as you mentioned, the statistical analysis found the relationship, the relationship probably does not mean the mechanisms. Thus the deeper mechanistic explanation was added in the revised version accompanied with the schematic diagram (Response to Q2 to Q3).

For the original data, we also submitted it as supplementary materials. Unfortunately, this data could not be found in the TC Discussions. We submitted it again and provide the link of the data as below:

"All the original data were available on the website of The Cryosphere Discussions (http://editor.copernicus.org/index.php?\_mdl=msover\_md&\_jrl=25&\_lcm=oc73lcm74a&\_acm= get\_supplement\_file&\_ms=50278&id=704538&salt=1523858444357408567)."

7. With the present analyses I would not recommend that this paper is published in The Cryosphere. The statistical analyses are limited and seem more spurious than hypothesis-driven. While an extensive dataset is available, I find that the authors provide little conclusions or results to significantly increase understanding of SOM accumulation or development in the north

Qinghai-Tibet Plateau. I recommend that the authors take the opportunity to re-examine their extensive dataset and look critically at which landscape processes have led to the interesting SOM dynamics we see in these boreholes.

**Response:** Thanks very much for the comment. We have revised it substantially, changed the title of the manuscript, and discussed the roles of landscape processes in the SOM accumulation and preservation in the mountain permafrost regions.